# Improving the Learning in Life Education for Young Children Aged 3 to 6 Years: A Review on the Research Themes of Early Childhood Life Education in Taiwan

**DOI:** 10.3390/children9101538

**Published:** 2022-10-09

**Authors:** Yi-Huang Shih

**Affiliations:** Department of Early Childhood Education and Care, Minghsin University of Science and Technology, Hsinchu 30401, Taiwan; shih262@gmail.com; Tel.: +886-03-0915306690

**Keywords:** early childhood life education, psychological wellbeing, review, young children, systematic review registration

## Abstract

The National Digital Library of Theses and Dissertations and the NCL (National Central Library) Taiwan Periodical Literature database were used to analyze dissertations and journal articles on early childhood life education. The researcher explored the research themes of life education for young children aged 3 to 6 years in Taiwan and offered strategies for improving learning in early childhood life education. Eight categories of themes were examined. In addition, the curriculum development of learning in early childhood life education was identified: (1) life event core curriculum, (2) sympathy core curriculum, (3) example core curriculum, and (4) anima core curriculum. Several teaching approaches of learning in early childhood life education are proposed: (1) teaching through picture books, (2) learning through experience, (3) teaching through instructional media, (4) teaching through concept mapping, (5) teaching through stories, and (6) teaching through parent–teacher collaboration. Developing and implementing the strategies of learning in early childhood life education is crucial for cultivating psychological wellbeing in young children and improving the quality of Taiwan’s early childhood education system.

## 1. Introduction

### 1.1. Life Education in an Early Childhood Development to Cultivate Psychological Wellbeing in Young Children Aged 3 to 6 Years

Psychological wellbeing in young children includes both their mental health and emotional health. Good psychological wellbeing gives young children the best chance to develop into healthy adults who have the coping skills in place to deal with day-to-day life. Good mental health and emotional health help young children develop socially, emotionally, mentally, and physically. Furthermore, loving relationships are key to young children’s mental health and emotional health. Learning to manage feelings is important to mental health and emotional health [1,2]. 

Each child is different in terms of height, weight, body build, and developmental stage. Early childhood is a critical period for individual growth and development, with experiences in early childhood affecting an individuals’ life situation in the future. The various cognitive processes of young children develop gradually throughout their various developmental stages. For example, critical thinking is a complex mental activity that develops relatively late during an individual’s mental development and is based on mental processes such as feeling, perception, and memory. Thinking skills develop along with cognitive processes during a child’s growth and learning [3,4,5]. Piaget described the different stages of cognitive development, noting that at each stage, children develop a new manner of functioning, thinking, and responding to their external environment [6,7].

Young children aged 3 to 6 years are in what Piaget calls the preoperational stage, also known as the egocentric or representational activity stage, which is a transitional stage between the sensorimotor stage and concrete operational stage. The so-called egocentrism refers to when preschool children perceive a social situation from their own point of view to the exclusion of the viewpoints of others. In addition, young children are unable to perform higher-level thinking as part of their cognitive processes, which increases the difficulty of implementing life education. Life education must, therefore, focus on the levels of feeling, action, and emotion, and cultivate psychological wellbeing in young children [5,6,7,8,9]. This study offers useful strategies for developing, improving, and implementing life education in an early childhood context, specifically, for young children aged 3 to 6 years. These recommendations can be used to assist young children’s development through life education and cultivate psychological wellbeing in young children aged 3 to 6 years.

### 1.2. Conceptualization of Early Childhood Life Education

With science and technology increasingly integrated into daily life, many people pay too much attention to technological devices and ignore the human aspect. After humans evolved to attach importance to utilitarianism, education was tainted with a utilitarian atmosphere. Furthermore, an education system that places too much emphasis on the practicality of science and technology produces negative effects in postmodern society. These effects manifest as rising suicide rates among teenagers and middle-aged people and a disregard for one’s own life and the lives of others. Thus, advocating for life education has become even more crucial in Taiwan. Life education for young children focuses on how young children live in the world and on the type of world they must establish in the future to benefit both individuals and society at large [10,11,12,13]. 

Based on the literature analysis [14,15,16], this study revealed that life education for young children is an essential element of education in Taiwan. This education is centred on educating young children to love, respect, and care for themselves, other people, and the environment and on deepening the understanding of the meaning and value of life among young children. Life education assists young children in developing kindness and sympathy for humans and other living creatures and cultivate psychological wellbeing in young children [10,12,13,17]. 

### 1.3. Early Childhood as a Critical Period for the Implementation of Life Education to Cultivate Psychological Wellbeing in Young Children

Early childhood is the critical period for implementing life education to cultivate psychological wellbeing in young children. The inquisitiveness exhibited in early childhood represents the beginning of young children’s understanding of themselves and the people and objects around them; hence, at this time, young children begin to construct a meaning of life [2,18,19,20,21]. Huang et al. (2011) argued that to implement life education in early childhood, preschool teachers must design effective learning activities that incorporate early childhood life education and develop professional learning in early childhood life education to enable young children to deepen their life concepts and cultivate psychological wellbeing in young children [11,12,22].

In sum, life education courses assist in preparing young children to enter primary school, where they can further develop life concepts such as living in harmony with others and cultivate psychological wellbeing in young children. Therefore, preschool teachers must design comprehensive life education courses for implementation during early childhood [11,12,22].

The key goals of life education for young children are to cultivate children’s love for themselves, other people, and the environment. Early childhood is a critical period for life education. Early childhood life education has been undervalued in preschools, resulting in gaps in the Taiwanese preschool curriculum. However, research literature in early childhood life education is now present in Taiwan (Table A1, Appendix A). The focus of this study was developing and improving life education curricula, which can help young children understand and prepare for life and ensure their psychological wellbeing. Because of the state of the research on early childhood life education in Taiwan, a review of the life education literature was not conducted. The researcher instead reviewed the literature on learning through early childhood life education to investigate approaches that could benefit the development and implementation of such curricula. 

## 2. Research Method

The researcher employed documentary analysis to examine studies related to life education for young children, explore key themes, and develop suggestions for life education [12,23,24]; eight themes were identified (Table 1).

Most research studying early childhood life education has focused on picture books. The least-studied topic is the characteristics and content of young children’s life education. The researcher consulted the National Digital Library of Theses and Dissertations in Taiwan and the NCL (National Central Library) Taiwan Periodical Literature database and extracted 29 dissertations and 13 journal articles, respectively, related to early-childhood life education for analysis (Table A1; Appendix A). These materials were reviewed to deepen our understanding of the topic and facilitate a structured analysis.

## 3. Research Themes in Early Childhood Life Education in Taiwan

Table A1 presents the literature. This study extended this research through elaborating on eight key themes.

### 3.1. Characteristics and Content of Young Children’s Life Education

The characteristics and content of young children’s life education.

#### 3.1.1. Results

Shih et al. identified the following characteristics of young children’s life education. (1) Diversity: Children’s life education covers multiple aspects of life and death, cultivating children’s self-knowledge, respect for themselves and others, empathy, and respect for life, plants, and animals. (2) Human nature as a whole: Life education for young children represents ‘whole-person education’, emphasizing balanced relationships between children and the self, other people, and objects and includes the aspects of knowledge, emotion, and will, with the goal of enhancing children’s life practice ability. (3) Individuality: Children’s life education promotes each child’s physical and mental health, beneficial living habits, interpersonal relationships, and understanding of ethical concepts. (4) Positivity: Early childhood life education assists children in accepting the unchangeable facts of life and overcoming difficulties in life, cultivating children’s positive attitudes in the face of challenges, and promoting their senses of self-worth and self-affirmation. Through life education, children learn how to care for themselves, for others, and for the natural environment [25]. 

Shih and Wu further investigated the content of life education for young children, detailing children’s understanding of life, the origin of life, and life care, development, and practice [26]. 

#### 3.1.2. Discussion

In conclusion, Shih et al. and Shih and Wu have focused on the characteristics and content of young children’s life education. Through referencing this work, preschool teachers can understand the essence of early childhood life education and practice effective teaching [25,26].

### 3.2. Planting and Raising Activities in Early Childhood Life Education

Raising plants and animals.

#### 3.2.1. Results

Hsiau used the action research method to explore the process of implementing life education teaching in preschools. The main teaching design involved planting and raising activities. The research revealed that (1) planting and raising activities cultivate children’s attitude of respect for life; (2) planting and raising activities have a persistent effect on children’s life experience; and (3) demonstrating adult behaviours and attitudes in planting and raising activities is essential [14]. Li et al. examined the strategy of raising butterflies as part of preschool life education, and we further investigated the influence this activity had on children. The children learnt about the value of life and how to interact with butterflies through the activity, which was demonstrated by the teachers. The teachers guided the children to observe, take care of, and feed the butterflies. The children’s ideas about life were developed by performing the activity themselves and interacting with the insects, gaining an understanding of their responsibility to protect them. During this process the children developed an understanding and appreciation of the following concepts: (1) respect life in all its forms, (2) respect the environment, (3) feel a sense of responsibility through caring for the butterflies, (4) understand the natural life and death cycle, and (5) grow and change personal learning attitudes [27].

Wu also conducted action research in life education using insects for teaching and drew the following conclusions: (1) Insect activity courses can be used to achieve life education goals. (2) Through observation and feeding activities, positive attitudes towards insect life can be cultivated in children. (3) Through an insect-themed course, children’s respect, love, and care for insects can be used to overcome any fear they may have for the insects. (4) Children’s interest in the observation, feeding, and recording of the insects can be increased and maintained. (5) An insect-themed course can transform children’s attitude of indifference into one of love and care for the insects, an attitude which the children can then advocate to other people [28].

#### 3.2.2. Discussion

In conclusion, Hsiau (2002), Li et al. (2010), and Wu (2021) have comprehensively studied the theme of planting and raising activities. Such activities allow children to experience life in its various forms. Through this process, young children can be encouraged to care for the natural environment and the lives of animals [14,27,28].

### 3.3. Application of Picture Books in Life Education

Picture books.

#### 3.3.1. Results

Huang applied the action research method to integrate the use of picture books in a set of curricula for early childhood life education. The study results revealed that the picture book curricula were suitable for learning. Changes observed in the children included a shift in orientation away from lying and evasion towards honest and responsible behaviour and away from self-centredness and interpersonal conflict towards sharing with and assisting others [29]. Chen implemented an action research approach to explore story discussions based on picture books among young children, reporting that children’s life attitudes and behaviours were enhanced in terms of sharing, caring, and gratitude [30]. Chang introduced a picture book curriculum to increase young children’s understanding of their origins and uniqueness. Through picture books, young children learnt about respecting friends, cherishing siblings, caring for animals, appreciating life, and recognising the value of plants [31].

Wu and Wei selected eight early childhood life education picture books and analysed their content and teaching targets in conjunction with the teaching experiences of early childhood life education educators to develop a more holistic method for promoting children’s development [16].Wang adopted the action research method in the design and implementation of a ‘people and self’ life education course promoting self-esteem among children through an online picture book reading club. Wang offered the following research conclusions. (1) The reading club and self-esteem course experience promoted the children’s self-esteem development. (2) The ‘people and self’ life education picture book was the cornerstone of the children’s self-esteem learning. (3) The course improved the children’s self-esteem, respect for others, and consideration and care for others [32]. 

Pan used collaborative action research to examine the picture book-based teaching process for young children’s life education, measuring the changes in children and teachers after teaching. Pan determined that the transformation of young children through teaching with picture books included the following three aspects: (1) the ‘self–self relationship’, including understanding the origin of life, having the ability to self-affirm, and possessing a correct understanding of death; (2) the ‘human–self relationship’, including expressions of caring and concrete actions, love for family members, and a willingness to express ideas and appreciation; and (3) the ‘environment–self relationship’, including the attitude of respecting life and the establishment of the concept of environmental protection [33].

Li selected 100 picture books related to children’s life education based on their applicability to the physical and mental development and learning characteristics of children aged 2 to 6 years. They used the literary elements of the picture book, such as the theme, plot, characters, background, words, and pictures, as the basis of a four-dimensional connotation analysis of children’s life education. In addition, a database of 100 picture books for young children’s life education was established to serve as a reference for preschool life education [34].

Yuan applied the action research method to improve young children’s learning effectiveness and cognition in terms of respect and affirmation, love and caring, acceptance and giving, and cherishing and gratitude using picture books [35]. Lai determined that teaching with life education picture books increased children’s self-confidence, courage, and life attitudes, such as being appreciative, cherishing material resources, and exhibiting adaptability [36]. 

Fan incorporated life education into picture book teaching to cultivate preschool children’s awareness of love and respect. Three key learning perspectives must be explored, namely, interaction with the self, interpersonal interaction, and interaction with the environment. After children’s abilities and learning are aligned, the main goals can be established. The next step is to select the children from the class who meet the age-based physical and mental development criteria suitable for measuring the chosen learning indicators, integrating the use of picture books into the curriculum design [37].

#### 3.3.2. Discussion

The form of action research, with picture books, is widely used. Picture books have been demonstrated to be effective teaching materials for young children’s life education. Huang, Chen, Pan, Li, Yuan, and Lai all focused on integrating picture books into life education. In addition to the use of picture books, teachers integrate a ‘teach-by-example’ approach into children’s life education curricula [29,30,33,34,35,36].

### 3.4. Interaction between Teachers and Students and Teachers’ Attitude towards Life

Interaction between teachers and students and teachers’ attitudes towards life.

#### 3.4.1. Results

Wang studied preschool life education practices and the exchange of life experiences in the interaction between teachers and children. They determined that life education is a process in which life affects life, such that an adult’s life experience can influence a child’s life experience. This interaction is crucial for developing children’s attitude to life and empathy for others and must, thus, be implemented in early childhood education [15]. 

Ye led a group of preschool teachers in a life exploration course, after which several improvements in the teachers’ implementation of life education were noted. Compared with before the course, the teachers faced life more actively in general, exhibited more kindness when interacting with their young students, and had a stronger grasp of the teaching objectives and curriculum design. They could also readily apply the activities practised in the life exploration course in their actual teaching [38].Chen described preschool teachers holding birthday activities for young children, which can assist both preschool teachers and young children in understanding life meanings [18]. Huang, Wei, and Hung explored preschool teachers’ attitudes towards life education. The findings of their quantitative study indicated that, in regard to life education, older teachers had more positive attitudes than younger teacher [22].

#### 3.4.2. Discussion

The aforementioned studies examined the interaction between parents and students, parents and teachers’ attitudes towards life, the practice of life education in Montessori preschools, and the difficulties and effects of preschool teachers’ implementation of life education for young children. This body of research demonstrated that the more favourable parents’ attitude towards life education for preschool children is, the higher their demand for life education is. Therefore, preschool teachers can promote life education and its crucial role in child development to parents. Teachers can further cooperate with parents in promoting life education and developing various teaching methods.

### 3.5. Parenting Attitudes and Teachers’ Current Implementation of Life Education

Parenting attitudes and teachers’ life education practices.

#### 3.5.1. Results

Wang studied the relationship between parents’ attitudes towards life education and their perception of need among preschool children in Kaohsiung City. The study results revealed that (1) the parents’ attitudes towards life education and parenting of preschool children were at an upper level; (2) the parents’ perceived need for life education for preschool children was at an upper level; and (3) the parents in this study were more likely to be women, be married, have a college-level education or above, have had some exposure to death and funerary experiences, and exhibit a life education attitude consistent with their parental attitude. The more consistent the life education and parenting attitudes of the parents were, the greater was their desire for life education for preschool children [39].

Li examined the life attitudes of preschool teachers in Changhua County, the implementation of life education, and the relationship between the two, revealing the following results: (1) Teachers’ attitude towards life was positive, and they felt the strongest about the ‘love and caring’ aspect. (2) The top five methods of implementing life education in preschools were picture book teaching and opportunity education, planting and raising, oral lecturing, and parent–child education activities. (3) The top five themes in life education courses implemented in preschools were ‘knowing and caring for one’s own body’, ‘experience sharing and others’ happiness’, ‘experience with and respect for others’ feelings’, ‘cultivating good living habits’, and ‘learning group cooperation and practising mutual support’. (4) Among teachers’ attitudes towards life, the stronger the attitude of love and caring was, the higher was the proportion of life education implemented in their curricula. (5) The more positive the teachers’ attitude towards life was, the more positive their attitude towards children’s life education was [40].

Tsai discussed the current situation and effectiveness of, and obstacles to, implementing life education for young children in Yilan County, focusing on differences in the effectiveness of implementation by preschool teachers of different backgrounds. The research revealed the following results. (1) The five most common implementation methods of life education employed by the preschool teachers were picture book teaching, opportunity education, narration, planting and raising, and parent–child education activities. (2) The top five life education course themes were ‘knowing and caring for one’s body’, ‘cultivating good living habits’, ‘caring for the natural environment’, ‘experiencing the joy of sharing with others’, and ‘affirming one’s self-confidence’. (3) The five most commonly employed information sources related to life education were reading book lectures, seminars, teaching observation, network system, and media [41].

Li explored the current situation and differences in the attitudes and practices of young children’s parents with respect to children’s life education in central Taiwan, analysing the predictive effects of the background variables on said attitudes and practices. Overall, the attitudes of the parents of the young children towards children’s life education were positive, supportive, and agreeable. Li argued that children’s practice of life education can be promoted by their parents and embodied in children’s everyday life experience, with the interpersonal and environmental dimensions requiring particular strengthening [42]. 

#### 3.5.2. Discussion

In conclusion, Wang, Li, Tsai, and Li have all investigated the themes of parental attitudes and teachers’ current implementation of life education. Parental attitudes towards and perceptions of the need for life education affect the practice of early childhood life education. Hence, the association between parents and teachers plays a key role in life education [39,40,41,42].

### 3.6. Multimedia Teaching, Concept Mapping, and Fables for Early Childhood Life Education

Multimedia education, concept mapping, and storytelling.

#### 3.6.1. Results

Pan conducted research on of the multimedia teaching of self-concept in young children’s life education. The production of multimedia teaching materials for young children has become a means through which children’s main life themes can be displayed [21].

Hsieh investigated the integration of film teaching, and concept mapping into young children’s life education. The research results indicated the following. (1) Selecting films that are consistent with the teaching objectives and children’s cognitive abilities and learning interests enhances children’s understanding of the film and its life meaning. (2) Film teaching can be integrated into young children’s life education to improve children’s cognition and life practices. (3) Through demonstration, teachers can guide young children in the concept mapping of film stories to assist children in learning life concepts. (4) The integration of concept mapping into young children enhances children’s understanding of respect for themselves and others, and appreciation [43].

Tang’s study of children in a mixed-age class in a preschool in New Taipei City applied the Zhuangzi (莊子) fables as a teaching material in children’s life education. The study indicated that young children enjoyed reading the Zhuangzi fables and actively participated in the follow-up learning activities; thus, implementing Zhuangzi as a teaching material in young children’s life education is feasible. Traditional Eastern classics can evoke interest and a willingness to learn. The life philosophy contained therein teaches young children about the self, interpersonal relationships, and core cultural aspects and can be applied to modern life [44].

#### 3.6.2. Discussion

Moreover, multimedia teaching materials serve as an effective platform through which to introduce relevant life themes to young children. Through appropriate strategies, film teaching can be integrated into children’s life education to improve children’s awareness and life practices, and concept mapping and experiential activities can be applied to develop children’s cognition and behaviours in terms of love and caring, respect for themselves and others, and cherishing happiness. Furthermore, implementing fables and storytelling in children’s life education teaching is also feasible.

### 3.7. Implementation Methods and Effects of Experiential Activities in Early Childhood Life Education Courses

Experiential activities and their effects.

#### 3.7.1. Results

Deng stated that experiential activity is the core of experiential learning, and investigated the implementation of life education teaching activities for young children to understand the effect on children’s performance and development. The study revealed that multiple teaching methods can be used to cultivate children’s ability to observe, think, and interact with others and to improve behaviours related to care, gratitude, and repayment and attitudes of respect and courteousness [45]. Pan concludes that experiential activities are very important in planning activities, and researched young children’s life education from the perspective of experiential activities, reporting the following results: (1) follow-up extension activities must be incorporated when designing experiential activities, (2) early childhood is a suitable developmental stage for life education, (3) parents as education resources are essential to the success of teaching, (4) life education themes, especially death education, must be taught at the appropriate time point, (5) effective methods for teaching life education to young children include increasing motivation through stories and applying experiential activities as teaching strategies, and (6) experiential activities such as raising animals and caring for plants are effective activities for children’s life education [46].

Liao implemented life education in a mixed-age preschool class of 4-to-6-year-old children using an immersive thematic curriculum to increase the children’s understanding of communication between people, people and the environment, people and the universe, and themselves and others. This research indicated that the children exhibited positive self-development, a more loving attitude towards the environment and other living creatures, and a calmer attitude towards death after implementation of the immersive thematic curriculum [47].

Wu designed a set of activity plans integrating a walking tour around a neighbourhood and an experiential activity into young children’s life education, reporting improvements in children’s awareness and practice of life education, the connection between children’s feelings and experiences, their understanding of their living environment, and their learning experiences. The young children’s cognition and practice of the life education concepts of care, respect, and love exhibited positive development [48].

Fang used schoolground nature experiential activities to explore young children’s life education. The research results revealed the following: (1) the use of campus-based nature experiential activities is suitable for developing young children’s life education courses, (2) discussion and reflection after the nature experiential activities can further strengthen young children’s emotive responses in life education, and (3) the implementation of campus-based nature experiential activities can enhance young children’s life attitudes of respecting themselves, caring for others, and cherishing nature [49].

Tai designed activities integrating a hiking experience in Zhishanyan to conduct action research on life education. The combination of the hiking experience with children’s life experience education allowed the children to make connections between their feelings and experiences, enjoy an enriching learning experience, deepen their understanding of the community environment, and gain life knowledge [50].

Shih proposed the following methods of promoting the life education of young children: (1) using picture books; (2) discussing life situations with children; (3) allowing children to appreciate life education videos; (4) using nature as a teaching tool; (5) introducing children to the experience of raising small animals; and (6) cultivating children’s gratitude to others [11].

Lu conducted action research on the integration of community tour experiential activities into life education. The study demonstrated that the combination of community resources with excursion-based experiential activities in children’s life education is an effective approach that improves children’s participation and positive feedback and increases parental support and education. This combination approach can, thus, feature as a core component of preschool life education courses. The incorporation of community tour experiential activities into the life education curriculum can encourage children to respect life and cherish natural resources, deepening their understanding of the environment in which they are growing up. This can assist in generating a sense of identity in which children internalise these life values [51].

Shih studied the implementation of life education for infants (aged 0–3 years) to identify the optimal approach for infant life education. The following techniques were employed in infant life education: (1) encouraging embracing among the infants, (2) calmly accepting undisciplined behaviour, (3) encouraging and praising, (4) smiling at the children, (5) playing many games, (6) using appropriate body contact, and (7) introducing the infants to nature [52].

Chan explored the implementation of a preschool experience course focused on playing with nature and integrating ecological awareness into life education to enhance children’s respect for the natural environment, their knowledge of insect conservation and plant ecology, and their protective practices towards the ecological environment. The results of life education course are aligned with the goals of early childhood education and can promote children’s respect for life [53]. 

#### 3.7.2. Discussion

As discussed in relation to the aforementioned research, using experiential activities as a teaching strategy for conducting life education for young children is an effective method. When natural experience is integrated into life education, children’s empathy and caring behaviours increase significantly. Therefore, preschool teachers can develop experiential teaching methods to enhance children’s life attitudes in terms of self-respect, caring for others, and cherishing nature. 

### 3.8. Philosophy of Early Childhood Life Education

The philosophy of early childhood life education.

#### 3.8.1. Results 

Chang examined the basic concepts of life education from the philosophical standpoint of Friedrich Froebel, known as the ‘Father of Early Education’, with a particular emphasis on life education. Froebel’s philosophy contains many concepts related to life education, including a belief in the close relationships of nature, humans, and God and the concept that children have natural ‘gifts’ that must be unfolded. Chang recommended that life education curricula in preschools communicate to children that everything contains the nature of God, provide children with opportunities to be close to nature, emphasise play-based activities, offer children real work experience, promote the creation of favourable habits, encourage viewing situations from different perspectives, and increase children’s understanding of the relationship between parts and the whole [54].

Chen investigated the themes of life and death as explored in adult–child dialogue to propose effective methods for implementing life and death education. In Chen’s study, the child’s conceptual constructs were shaped through the adult’s response to their questions, thus framing the child’s relation to the world [55]. 

Chang explored the act of caring being explicitly demonstrated by teachers and analysed four dimensions of the implementation of the ethics of care, namely modelling, dialogue, practice, and confirmation, which can serve as a reference for life education. Preschool teachers must integrate the ideas of the ethics of care into life education methods and practices [56].

Chen et al. analysed that young children must enjoy a reasonable range of freedom in education to enable them to develop an independent personality. Young children can remove obstacles to their life development using their own free will. Moreover, with such freedom, young children can conform to the natural laws of development and habituate correct and meaningful actions [57]. 

#### 3.8.2. Discussion

The researcher analysed four articles (Chang, Chang, Chen, Chen et al.) to examine early childhood life education using perspectives on philosophy, including those of Maria Montessori and Friedrich Froebel, and the ethics of care. Another study conducted life education centred on life and death themes using dialogue and problem-posing strategies. However, this research on early childhood life education has left little room for alternative perspectives, such as that of educational philosophy. For example, the educational philosophies of Rudolf Steiner and John Dewey have not been explored in the context of early childhood life education [54,55,56,57].

## 4. Improving the Learning in Early Childhood Life Education of Taiwan’s Education System 

Among the themes, raising plants and animals was particularly crucial because it provides children with opportunities to various aspects of life. Therefore, animals and other living beings should be incorporated into the curriculum. Picture books, concept mapping, instructional media, and stories are also effective in life education. Parental attitudes towards and perceptions of life education affect the practice of life education. Teachers can promote life education among parents through parent–teacher associations. Experiential activities are also effective in life education because they help children develop sympathy and care, which are crucial to life education. Adults’ behaviour should also be incorporated into the curriculum because it sets an example for children and teaches them to value care and to care for others and the environment. 

### 4.1. Strategies for the Implementation of Preschool Curricula

This study developed several strategies in curricula based on the results.

#### 4.1.1. Life Event Core Curriculum

Teachers should select courses that children are familiar with so that the courses resonate with children’s life experiences [23]. Life education is not an educational movement or trend but an essential life course. The life event core curriculum is a curriculum designed on the basis of a care-related life event. Therefore, life education must be continually implemented in all teaching activities or life counselling sessions, and the practices learnt therein implemented in life [15]. Furthermore, specific situational life events can be used as subjects to teach children about life practices [58]. These themes and events from daily life must be shared and discussed with young children to allow them to have more profound life experiences relating to these events and to achieve the greatest teaching effect. For example, if a child’s grandparent dies, preschool teachers can use the opportunity provided through this life event to explain the concept of death and how to seek comfort from one’s family during difficult times [14]. Another such opportunity is provided through children’s activities based on caring for flowers and plants. If the flowers and plants wither, preschool teachers can explain these flowers and plants are part of the cycle of life and death, as is the case for all products of nature, including people. 

Through such teaching, young children can deepen their understanding of death in a calm and reassuring setting. Conversely, Wang and Huang discussed the traditional Chinese approach of protecting children from death, even to the point where parents cover their children’s eyes when passing a funeral service. However, adults cannot fully shield children from learning about death, with such actions potentially harming rather than protecting children. Death is an integral part of life. Wang and Huang studied preschool-aged children who had witnessed a dog dying in a car accident, an often upsetting life event, and investigated how the children processed and understood this death. The researchers suggested the following to parents: (1) allow children to learn about the life cycle in a natural manner, (2) have a positive attitude about life, (3) provide comfort when children are exposed to death, (4) and choose television programmes with death-related content cautiously. They also made the following suggestions to teachers: (1) enrich life education curricula, and (2) teach children to express their feelings and emotions. This research can assist parents and teachers in understanding how children process the concept of death and how adults can support this conceptual process [59].

The Early Childhood Education and Care Curriculum Framework (Ministry of Education) described how teachers can expand children’s life experience, provide opportunities for children to share their own life experience, and encourage children to recognize their family’s love and care towards them, particularly in cultural contexts. Various tools, such as language, pictures, body movements, and role-playing, can assist children in articulating their experiences and feelings and provide opportunities for them to listen to the experiences of others, encouraging children to explore their living environment and interpersonal relationships. Furthermore, preschool teachers can use daily life events as educational opportunities [13,17].

#### 4.1.2. Sympathy Core Curriculum

Moral sentiments are innate in human beings, including that of sympathy [60]. Etymologically, the English word ‘sympathy’, composed of ‘sym’ and ‘pathy’, is rooted in the Greek ‘sympatheia’ or Latin ‘sympathia’. ‘Sym’ is derived from ‘syn’, which means together, and ‘pathy’ means emotion and suffering. Therefore, empathy refers to the state of being commonly affected and feeling the same as others, or the ability to produce such a feeling state. However, if ‘pathy’ means pain, ‘sympathy’ represents pity, with the sympathiser feeling the pain state of another [61].

The foundation of life education for young children lies in people’s emotional experience of sympathy. Preschool teachers must develop life education courses that allow and encourage young children to experience sympathy through experiential activities, enabling children to feel emotions directed towards others. Such experiential activities expand young children’s emotions and deepen children’s sympathy for others who are different from themselves, which is an essential part of life education curricula. Therefore, when teachers design life education courses, they can place empathy at the core of the curriculum and apply picture books to enhance young children’s emotional identification and development [62]. For example, preschool teachers can use the picture books Inaudible Concert, Super Brother, Me and the Wild Dogs Near My Home, and Weiwei Looking for Memories to discuss the meaning of life and cultivate empathy among young children.

#### 4.1.3. Example Core Curriculum

The practice is driven by example set by preschool teachers and parents, a statement supported by Pan [21]. The more positive the life attitude of preschool teachers is, the more positive their attitude towards the life education of young children is (Li, 2011) [40]. Chen observed that, in raising and planting activities, demonstrations of adult behaviour and attitude are crucial [30]. Taiwan’s Ministry of Education also advocated for preschool teachers leading by example and demonstrating the behaviours of sharing, caring, communication, and listening in life education teaching. If adults demonstrate behaviours that contradict the concept of caring about people and their environment, how can children be asked to care? Adults must ensure that they are not setting an unfavourable example for children [17]. Noddings described teaching by example as the performance involved in the teacher education profession. Teaching by example, based on caring ethics, strengthens the moral foundation of human relationships as part of the inevitable process of human growth [63,64].

Teaching by example does not rely on telling children to care but demonstrates and generates reciprocal caring relationships through the experience of being provided care [63,64,65]. Therefore, teaching by example is the key to early childhood life education practice based on a sympathy core curriculum [30].

#### 4.1.4. Anima Core Curriculum

All living beings collectively form the living world and life itself. Life entails a continuous series of changes and is a cumulative process [22]. Early childhood life education is intended to develop children’s life by enacting a process in which life affects life [15]. To establish an anima core life education curriculum for young children, teachers must recognize that children are living individuals who can be influenced by the teachers’ lived experience. Teachers must, therefore, determine what aspects are imperative for inclusion in their curriculum for young children. Young children tend to learn life education more effectively in humanistic learning environments with anima at their core. The lives of teachers and of children merge, allowing teachers and children to forge mutual life connections and meaningful relationships. 

### 4.2. Strategies for the Implementation of Teaching Methods

This study developed several recommendations for teaching strategies based on the results.

#### 4.2.1. Teaching through Picture Books

The use of pictures and texts in picture books allows children to imagine the story development. Picture books also have the functions of providing knowledge, transferring concepts, and guiding emotions [12,14]. Children’s life education courses designed with picture books are regarded as particularly suitable for preschool children’s learning. Some studies (Huang, Shih) have employed picture book teaching to stimulate children’s sympathy through appropriate narratives. When children identify with the characters in picture books, they can feel sympathy and empathy when misfortune befalls these characters, which thereby produces or enhances the effect of life education [12,29]. Early childhood life education promotes the practice of life care [14,29,66]. 

#### 4.2.2. Learning through Experience

Life education can be learnt experientially, which can more deeply embed life education values in children than instructive teaching can. Practical and experiential activities, such as planting and raising, can enable children to recognize the preciousness of life and cultivate an attitude of respect for life. Through discussion and reflection following experiential activities, children’s positive feelings and attitudes towards life can be further strengthened [1,12,14,21,58,66,67].

The cognitive development of young children can be divided into the sensorimotor stage and preoperational stage based on Piaget’s theory of development [6,7]. At these stages, children are unable to perform higher-level cognitive thinking and cannot experience life as richly as adults. Therefore, experiential teaching is necessary to teach children about life. Raising animals and planting plants are suitable experiential activities for life education, allowing young children to encounter and understand the life and death cycle [12,21].

Wu argued that children can then share their experiences and reflections with others through drama performances (role-playing), thus extracting knowledge from experience and transforming said knowledge back into actions [28]. In addition, Hsieh asserted that experiential activities combine children’s feelings and actions into life experience, assisting them in internalising life education values. Therefore, preschool teachers are recommended to apply experiential activities in their life education teaching methods [43].

#### 4.2.3. Teaching through Instructional Media

The instructional media used in teaching can be divided into soft, hard, audio-visual, and communication media. With the advancement of technology and enrichment of instructional resources, the use of audio-visual instructional media in teaching has become a trend. Teachers select appropriate instructional media to assist teaching according to the characteristics of their subjects to achieve the optimal learning effect. Instructional media enhances children’s interest in learning, and the integration of film media into children’s life education can improve children’s cognition and attitudes. Children’s concept of death is often drawn from media. The selection and implementation of films that meet the teaching objectives, cognitive development goals, and children’s interests deepen children’s understanding of the films and their meaning [68,69].

Furthermore, providing a brief description increases children’s attention when viewing, their comprehension of complex plots points, and resonance with the film meaning. The posing of open-ended questions after film viewing further improves children’s understanding of the film content and meaning [12,14,21,43].

#### 4.2.4. Teaching through Concept Mapping

Concept mapping is a tool or strategy for presenting the relationships in a conceptual network. Developed by Novak and Gowin, this tool emphasises the hierarchy of concepts, concept nodes, and the relationship between concepts. Thus, a network graph of concept nodes and connections, which are represented using connecting lines in the graph, is generated. The concept map typically begins with a key concept (or central theme) out of which branches develop, eventually generating various concepts of a higher level (a more inclusive concept) or lower level (usually referring to factual information). The organisational graph of a conceptual hierarchy transforms from an abstract into a concrete form [43,70]. Teachers can use concept mapping for children’s life education to enable children to understand the concept of life, as illustrated in Figure 1.

The diagram presented in Figure 1 can be used to represent the concept of life, teaching children about their role and that of other people and nature. Children are then guided to treasure themselves, respect others, and cherish nature. We maintain that this type of concept mapping is suitable and effective for teaching children about the concept of life.

#### 4.2.5. Teaching through Stories 

A story conveys human experience. Parker noted that children favour fairy tales because these stories cushion the harshness of reality by introducing an alternative world driven by children’s imagination [71]. Children enjoy listening to stories, and the use of diverse stories can maintain their engagement and motivation to learn [12,21]. Tang reported that children enjoyed reading the Zhuangzi (莊子) fables and actively participated in follow-up learning activities. This demonstrated the feasibility of implementing the Zhuangzi fables and their life philosophy in children’s life education to deepen children’s thinking about the self, others, and their surroundings [44]. 

#### 4.2.6. Teaching through Parent–Teacher Association

Family education is an essential part of life education, preschool teachers must fully utilise parental resources [12,21]. Whether teachers can effectively use parental resources is key to the success or failure of early childhood life education [14,21,39]. 

Related research (Fang, 2012; Hsiau, 2002; Huang, 2004; Pan, 2006) has supported the crucial role of the family (parents) in life education [12,14,21,29,49]. The life education of young children must be implemented over a long time, with continuity and the cooperation of family (parents) to be successful [29]. Parents may have their own desires for preschool children’s life education, but only with the support and cooperation of parents can children’s life education be integrated into long-term practice.

On the basis of the analysed studies, we demonstrated that the optimal approach for young children’s life education involves both the efforts of teachers and cooperation of parents. Teachers can promote parent–teacher cooperation by encouraging parents to participate in certain life education courses. For example, parents and children can plant and observe the growth of flowers together. 

## 5. Limitation and Recommendation

This study only conducted a literature review and analysis on the research themes of early childhood life education in Taiwan. Therefore, the researcher could only assess the research context of life education in Taiwan. However, on the basis of the theme analysis, particular themes can be selected for further in-depth discussion in the future. Finally, the researcher recommends that a cross-country comparative study of early childhood life education be conducted to broaden our understanding of early childhood life education, and to construct a basic framework about life education for young children in international educational systems.

## 6. Reflection and Conclusions

### 6.1. Reflection

By emphasizing the essential role of life education, the researcher aims to promote the development of life education in preschools in this study. Ultimately, life education curriculum development and improvement can assist young children in understanding and preparing for various life situations and cultivate psychological wellbeing in young children aged 3 to 6 years.

The learning interest construct has been proposed over three decades. To date, learning interest is still an ongoing discussed topic in academic and in practice. Preschool teachers should begin to monitor young children’s learning process from early childhood, because their learning interest may start to decline from preschool [72,73]. The issue of life education for young children is topical and worth raising as too much emphasis can be placed on the implementation of a narrow academic curriculum from an early age. It is relevant because it links to questions of an education’s purpose, and how we do indeed create a world worth living in. It is also infused with values that may be culturally specific, which would be insightful to acknowledge and unpack. However, more importantly, preschool teachers need to think about how to develop interesting lessons and teaching to enhance children’s learning interest.

### 6.2. Conclusions

The National Digital Library of Theses and Dissertations and the NCL (National Central Library) Taiwan Periodical Literature database were used to analyse dissertations and journal articles on early childhood life education. The researcher explored the main themes of life education for young children in Taiwan. On the basis of the above inquiry, the study offered strategies for the practical implementation of curriculum-based professional learning in early childhood life education. The following themes were examined: (1) characteristics and content of young children’s life education; (2) planting and raising activities for early childhood life education; (3) application of picture books in young children’s life education; (4) interaction between teachers and students and teachers’ attitude towards life; (5) parental attitudes and teachers’ current implementation of life education; (6) multimedia teaching, concept mapping, and fables for early childhood education; (7) implementation methods and effects of experiential activities in early childhood life education courses; and (8) philosophy of early childhood life education.

The following four areas for curriculum development were identified: (1) life event core curriculum, (2) sympathy core curriculum, (3) example core curriculum, and (4) anima core curriculum. Several teaching approaches were proposed, namely (1) teaching through picture books, (2) learning through experience, (3) teaching through instructional media, (4) teaching through concept mapping, (5) teaching through stories, and (6) teaching through parent–teacher collaboration. 

Young children aged 3 to 6 years are in what Piaget calls the preoperational stage, also known as the egocentric or representational activity stage, which is a transitional stage between the sensorimotor stage and concrete operational stage. The so-called egocentrism refers to when preschool children perceive a social situation from their own point of view to the exclusion of the viewpoints of others. In addition, children are unable to perform higher-level thinking as part of their cognitive processes, which increases the difficulty of implementing life education. Life education must, therefore, focus on the levels of feeling, action, and emotion [5,6,7,8]. This study offers useful strategies for developing, improving, and implementing life education in an early childhood context. The exploration undertaken in this research can benefit the development of professional learning in early childhood life education in different countries and cultivate psychological wellbeing in young children aged 3 to 6 years.

Most studies have focused on picture books in early childhood life education, and few have explored the characteristics and content of young children’s life education, which, thus, warrants investigation. Three studies each have analysed raising plants and animals; multimedia education, concept mapping, and storytelling; and the philosophy of early childhood life education, which also require further investigation.

## Figures and Tables

**Figure 1 children-09-01538-f001:**
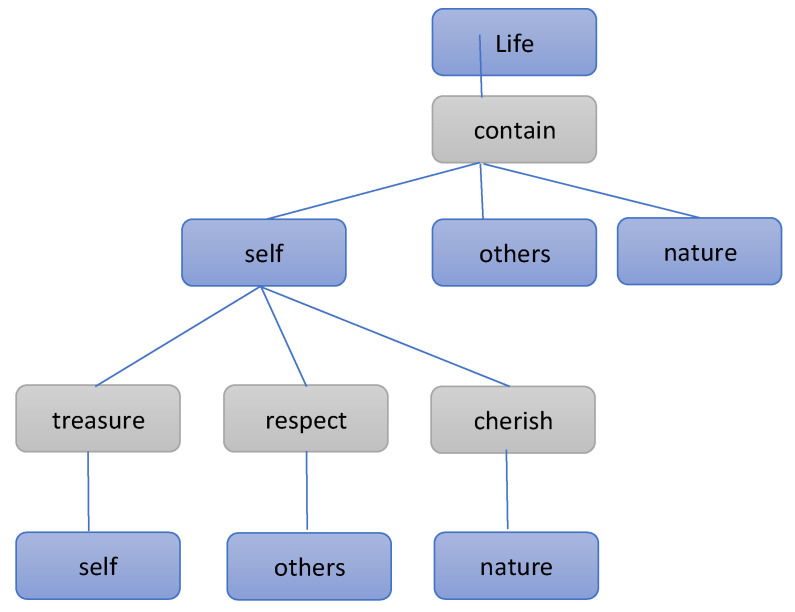
Life diagram. (Source: Developed in this study).

**Table 1 children-09-01538-t001:** Themes extracted from studies on life education for young children.

Serial Number	Theme	Number of Articles
1	Characteristics and content of young children’s life education	2
2	Raising plants and animals	3
3	Picture books	10
4	Interaction between teachers and students and teachers’ attitude towards life	4
5	Parental attitudes and teachers’ life education practices	4
6	Multimedia education, concept mapping, and storytelling	3
7	Experiential activities and their effects	12
8	Philosophy of early- childhood life education	3

(Source: Developed in this study).

## Data Availability

Not applicable.

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
