# Peer review of "Improving the Learning in Life Education for Young Children Aged 3 to 6 Years: A Review on the Research Themes of Early Childhood Life Education in Taiwan"

_children, 2022, doi:10.3390/children9101538_

Round 1
Reviewer 1 Report (New Reviewer)
dear author(s)
thank you for your very interesting article.
It is very nice to see how life education is addressed in Taiwan.
Nevertheless, there are some suggestions on how to improve the article and make it more reader-friendly:
1) It is not clear whether this concept is already implemented in Taiwanese curricula. You mention studies but do not address the state of pre-school education in Taiwan. You mention Confucianism and its values, but you never mention the implications for preschools. Is it not common for human values to be central to the curriculum? Is materialism predominant? If so, how does it manifest itself? In your introduction, please explain in more detail what is missing from the Taiwanese preschool curriculum.
2) What is the essence of 1.2 and 1.3. What do you want to draw attention to? What is missing in the Taiwanese curriculum that makes these ideas so important? They are very vague and general
3) 1.3 Typo in line 111. Lines 137 to 141 are actually the core of this study, but they are hidden in chapter 1.3. Why didn't you mention the aim of the study right at the beginning?
4) Table 1 is the core of this literature review. Yet, you do not include important information such as the sample and the type of preschool - public or private - since you mention Montessori later. Nor do you address how the Taiwanese curriculum is structured. Is life skill not part of it?
5) There is no introduction between chapters like 3.1. to 3.1.1. or 4.1 to 4.1.1. which makes it very difficult to know what to expect for the reader. I think a short introduction would be very useful.
6) I suggest reorganising chapters 3.1. to 4.6.5. as they are very long and your explanations of each study are also very long. I suggest shortening the chapters and also summarising them. For example, you could focus on what are the similarities or differences between the studies. For example: While Deng (2005) states that x,y, is the core of experiential learning, Pan (2006) concludes that x,y, are very important in planning activities. This way you get summarised texts that focus on the most important aspects and can be compared with other texts where these aspects are missing.
7) I have the impression that many ideas (as in 3.7.1 and 4.3.2 and 4.3.3 etc.) are repeated, but under different names. That is why it is important to be brief.
8) I have considered whether it is not better to focus only on 2-3 important aspects and not on all 6, as this overloads the text. In the end, readers don't know what the main idea is, what is important and what is not.
9) Finally, redo your methodology and the way you present the results. Be more concise, design a different structure and read other literature reviews to see how they do it.
Good luck
Author Response
|
The first reviewer’s comments
|
Modified page |
Revise
|
||
|
Thank you for your very interesting article. It is very nice to see how life education is addressed in Taiwan.
|
|
Thanks to the reviewer’s affirmation for the work a significant contribution to the field. |
||
|
It is not clear whether this concept is already implemented in Taiwanese curricula. You mention studies but do not address the state of pre-school education in Taiwan.
|
3
|
Modified according to the reviewer’s opinion. Modify the page as shown on page 3. Line 97-109.
|
||
|
You mention Confucianism and its values, but you never mention the implications for preschools. |
2
|
The author has deleted Confucianism to avoid many points in a paper. Modify the pages as shown on page 2. Line 61-79.
|
||
|
Is it not common for human values to be central to the curriculum? Is materialism predominant? If so, how does it manifest itself?
|
2
|
The author have deleted materialism to avoid many points in a paper. Modify the page1 as shown on page 2. Line 61-79.
|
||
|
In your introduction, please explain in more detail what is missing from the Taiwanese preschool curriculum.
|
3 |
Modified according to the reviewer’s opinion. Modify the page as shown on page 3. Line 97-109.
|
||
|
What is the essence of 1.2 and 1.3. What do you want to draw attention to?
|
|
The essence of 1.2 emphasized that early childhood is the critical period for implementing life education to cultivate psychological wellbeing in young children. The author draws attention to this.
The essence of 1.3 emphasized that life education in an early childhood development to cultivate psychological wellbeing in young children aged 3 to 6 Years. The author draws attention to this. |
||
|
What is missing in the Taiwanese curriculum that makes these ideas so important? They are very vague and general. |
3 |
Thanks to the reviewer’s opinion Modified according to the reviewer’s opinion. Modify the page as shown on page 3. Line 97-109.
|
||
|
1.3 Typo in line 111. Lines 137 to 141 are actually the core of this study, but they are hidden in chapter 1.3. Why didn't you mention the aim of the study right at the beginning? |
1 2
|
Thanks to the reviewer’s opinion Modified according to the reviewer’s opinion. Modify the page as shown on pages 1,2. Line 25-58.
|
||
|
Table 1 is the core of this literature review. Yet, you do not include important information such as the sample and the type of preschool - public or private - since you mention Montessori later. Nor do you address how the Taiwanese curriculum is structured. Is life skill not part of it?
|
|
Thanks to the reviewer’s opinion However, there are no the sample and the type of preschool - public or private in some research literature. So, It is difficult.
|
||
|
There is no introduction between chapters like 3.1. to 3.1.1. or 4.1 to 4.1.1. which makes it very difficult to know what to expect for the reader. I think a short introduction would be very useful.
|
8 9 10 11 12 |
Thanks to the reviewer’s opinion Modified according to the reviewer’s opinion. Modify the page as shown on pages 8, 9, 10, 11, 12. Line 131-132; Line 134. Line 159-160; Line 194. Line 256-257; Line 286-287. Line 338-339; Line 372-373. Line 455. Line 507; Line 601-602.
|
||
|
I suggest reorganising chapters 3.1. to 4.6.5. as they are very long and your explanations of each study are also very long. I suggest shortening the chapters and also summarising them. For example, you could focus on what are the similarities or differences between the studies. For example: While Deng (2005) states that x,y, is the core of experiential learning, Pan (2006) concludes that x,y, are very important in planning activities. This way you get summarised texts that focus on the most important aspects and can be compared with other texts where these aspects are missing.
|
12 |
Thanks to the reviewer’s opinion Modified according to the reviewer’s opinion. Modify the page as shown on page 12. Line 374-375.
|
||
|
I have considered whether it is not better to focus only on 2-3 important aspects and not on all 6, as this overloads the text. In the end, readers don't know what the main idea is, what is important and what is not. |
|
Thanks to the reviewer’s opinion If we only focus on two or three aspects, it may also lead to the failure to understand the situation of Taiwanese children's life education research.
|
||
|
Finally, redo your methodology and the way you present the results. Be more concise, design a different structure and read other literature reviews to see how they do it.
|
3 |
Thanks to the reviewer’s opinion Modified according to the reviewer’s opinion. Modify the page as shown on page 3. Line 97-117.
|

Reviewer 2 Report (New Reviewer)
This review is basic and easy to read.
My concern and comments have to do with the lack of description of the methodology used for the systematic analysis. This needs to be added.
Additionally, the step between the two parts (ch 3 and ch 4) is not described and needs to be clarified.
Good luck with the revision.
Author Response
|
The second reviewer’s comments
|
Modified page |
Revise
|
||
|
This review is basic and easy to read |
|
Thanks to the reviewer’s affirmation. |
||
|
My concern and comments have to do with the lack of description of the methodology used for the systematic analysis. This needs to be added. |
3
|
Modified according to the reviewer’s opinion. Modify the page as shown on pages 3, 4. Line 97-117.
|
||
|
Additionally, the step between the two parts (ch 3 and ch 4) is not described and needs to be clarified. |
14 15 |
Modified according to the reviewer’s opinion. Modify the page as shown on pages 8,14, 15. Line 130-131. Line 478-492.
|

Reviewer 3 Report (New Reviewer)
Title: The title is clear and concise, adequately indicating the core or essence of the study. The purpose and objectives of the study are clear.
Abstract: Justification for the study is well articulated.
Introduction: The premise of research is clearly formulated. The parameters of the research are clearly delineated.
Research Method: Research method is clearly explained. Procedures of data collection are extremely well documented.
Theoretical Framework: I applaud the author on the quality and selection of appropriate theories for this study ‘Life education’ (Friedrich Froebel) & Piaget’s theory of development (Piaget, 1970).
Literature Review: Key authors on the subject were consulted and enough sources were consulted.
Results and Discussion: Based on the findings of this study, the author is capable to pursue further research studies of this kind. The results are very interesting and insightful. The study is a good contribution and shall add value in a scholarly arena.
Reflection and conclusion: Concluding thoughts are clearly articulated and align with the aims and objectives of the study.
References: Correctly formatted, alphabetical order observed; however, No. 26 (Faulconbridge, J., Hunt, K., & Laffa, A. (2018), is misplaced. Needs attention.
Generally, good command of standard English without spelling mistakes. A well-focused and organized discussion. Approval for publication recommended after minor correction
Author Response
Author Response Letter
Dear Editor and Reviewer
The author of this manuscript (Manuscript ID: children-1925280) has modified this manuscript according to the reviewer’s comments.
I resubmit this manuscript.
Thank you for the reviewer’s comments.
The author’s response letter illustrates the revision of this manuscript.
|
Manuscript ID: children-1925280
|
||||
|
Improving the Learning in Life Education for Young Children Aged 3 to 6 Years: A Review on the Research Themes of Early Childhood Life Education in Taiwan
|
||||
|
The reviewer’s comments
|
Modified page |
Revise
|
||
|
Title: The title is clear and concise, adequately indicating the core or essence of the study. The purpose and objectives of the study are clear.
|
|
Thanks to the reviewer’s affirmation |
||
|
Abstract: Justification for the study is well articulated.
|
|
Thanks to the reviewer’s affirmation |
||
|
Introduction: The premise of research is clearly formulated. The parameters of the research are clearly delineated. |
|
Thanks to the reviewer’s affirmation |
||
|
|
|
|
||
|
Research Method: Research method is clearly explained. Procedures of data collection are extremely well documented.
|
|
Thanks to the reviewer’s affirmation |
||
|
Theoretical Framework: I applaud the author on the quality and selection of appropriate theories for this study ‘Life education’ (Friedrich Froebel) & Piaget’s theory of development (Piaget, 1970).
|
|
Thanks to the reviewer’s affirmation |
||
|
Literature Review: Key authors on the subject were consulted and enough sources were consulted.
|
|
Thanks to the reviewer’s affirmation |
||
|
Results and Discussion: Based on the findings of this study, the author is capable to pursue further research studies of this kind. The results are very interesting and insightful. The study is a good contribution and shall add value in a scholarly arena.
|
|
Thanks to the reviewer’s affirmation |
||
|
Reflection and conclusion: Concluding thoughts are clearly articulated and align with the aims and objectives of the study.
|
|
Thanks to the reviewer’s affirmation |
||
|
References: Correctly formatted, alphabetical order observed; however, No. 26 (Faulconbridge, J., Hunt, K., & Laffa, A. (2018), is misplaced. Needs attention. |
17 |
Modified according to the reviewer’s opinion. Modify the page as shown on page 17. Line 774-775.
|
||
|
Generally, good command of standard English without spelling mistakes. A well-focused and organized discussion. Approval for publication recommended after minor correction.
|
|
Thanks to the reviewer’s affirmation |
||
Thank you for the reviewer’s comments.

Round 2
Reviewer 1 Report (New Reviewer)
Dear author,
thank you again for the revision of the paper. It is unfortunately still too long and therefore I suggest that you move Table 2 Studies on life education for young children to the appendix of this paper. You can make a short note that the list of studies can be found in the appendix of the paper.
Perhaps you can also find a more concise way to present the studies? This will take some considerable amont of time!
the last paragraph of the conclusions needs an English proofreading.
Author Response
Author Response Letter
Dear Editor and Reviewer
The author of this manuscript (Manuscript ID: children-1925280) has modified this manuscript according to the reviewer’s comments.
I resubmit this manuscript.
Thank you for the editor’s and reviewer’s comments.
The author’s response letter illustrates the revision of this manuscript.
|
Manuscript ID: children-1925280
|
||||
|
Improving the Learning in Life Education for Young Children Aged 3 to 6 Years: A Review on the Research Themes of Early Childhood Life Education in Taiwan
|
||||
|
The reviewer’s comments
|
Modified page |
Revise
|
||
|
Thank you again for the revision of the paper. It is unfortunately still too long and therefore I suggest that you move Table 2 Studies on life education for young children to the appendix of this paper. You can make a short note that the list of studies can be found in the appendix of the paper. Perhaps you can also find a more concise way to present the studies? This will take some considerable amont of time!
|
20 21 22 23 24 25 26 |
Modified according to the reviewer’s opinion. Modify the page as shown on pages 20, 21, 22, 23, 24, 25, 26.
|
||
|
the last paragraph of the conclusions needs an English proofreading.
|
26
|
The author has conducted English proofreading . Modify the pages as shown on page 26. Line 729-733.
|
||
Thank you for the editor’s and reviewer’s comments.

This manuscript is a resubmission of an earlier submission. The following is a list of the peer review reports and author responses from that submission.
Round 1
Reviewer 1 Report
This is a long paper, and although there is no word limit for this journal, the length is not justified. A succint and synthesised discussion of the literature would provide a better structure for the paper and support a more focused and insightful discussion of key findings. I am not recommending publication at the moment, as the paper requires considerable work. However, I encourage the author to persist. I have provided some comments in the body of the paper. What follows are suggestions for improving the paper overall.
Firstly, the term 'life education' is not common or may have different connotations across cultural contexts. A definition of life education is needed at the beginning of the paper. It would be useful to explicitly place this in a Taiwanese context as this is where you are drawing your material from. There are various descriptors of what life education is at various points throughout the paper, draw these together to provide a common reference point for the discussion from the start.
The paper is often repetitive. For example, the justification for life education appears in a number of different places and guises. Like the definition of life education this could be brought together, synthesised and discussed early in the paper.
On page 4, you refer to Confucian ethics. This led me to want to know more about the link between Confucian ethics and the Taiwanese conceptualisation of life education in early childhood. This highlights an opportunity to unpack the cultural context of life education.
Research Method: This section requires you to outline your search terms; explain how you came to select the theses and papers that you included (for example, did you look at all that you found?); and explain that you confined your search to Taiwan (and explain why).
Presentation of findings: I found this confusing. First you discuss 8 key themes extracted from the literature (although I am not sure whether all of these are themes per se, rather than the topics addressed by the studies). Then you discuss 4 characteristics . Then 4 core curricula. More information is needed on how each of these has been derived and the relationship between each.
The literature review: The literature review itself resembles an annotated bibliography. It is this approach in particular that makes the paper overly long. Explaining each paper consecutively is not sufficient for a literature review, or indeed a systematic review. More rigour is needed in analysing the literature, moving away from just describing what each paper focuses on and says. What is needed is a more synthesised discussion in which key themes (rather than topics) are identified. We need to know how these themes are arrived at. Please note the previous comment about needing to explain how your findings have been derived.
I feel that this has the potential to develop into a very interesting paper, but at this stage it resembles the first collation of literature. The issue of life education for young children is topical and worth raising as too much emphasis can be placed on the implementation of a narrow academic curriculum from an early age. It is relevant because it links to questions of education's purpose, and how we do indeed create a world worth living in. It is also infused with values that may be culturally specific, which would be insightful to acknowledge and unpack.
I have included some comments in the paper itself which you may find useful.

Reviewer 2 Report
Dear Author,
Thank you for your manuscript. The topic is important and interesting. However, the style the paper written is recommended to be improved.
First, the paper is very long and repetitive. In the Introduction, the Author continuously comes to the same statement that improving early childhood education can contribute to better well-being in young children, but when reading, I was wondering all the time - why is this study so important and what exactly comes out of this extreme amount of the studies cited?
Also, please provide a short and clear study aim at the end of the Introduction.
I recommend shortening the paper and providing information in a more concentrated manner. Also, in Table 1, the main findings from referenced studies would be more informative rather than the topics. That would allow to avoid the description of the studies' findings in the Results and Discussion section and shorten the manuscript.
Next, the study methodology should be improved by answering the following questions:
- What was the period of studies involved?
- What were the inclusion/exclusion criteria for the studies to be involved? (language written, country, study design, etc.)?
Also, Figure 1 should be improved. The Figure should be more self-explanatory and independent of the information in the text. Please add full terms.
Reviewer 3 Report
The intention of the study is appreciated which addresses the importance of early childhood life education and youth psychological wellbeing. However, a proper systematic review is needed as commented below. A thorough proofreading is also required. While the topic is of value, the article has major issues that must be addressed.
1.
Table 1 (page 4) presenting the results of the study should be more comprehensive. It should contain columns with elements that gives strength to the study, e.g., type of evaluation, measuring aspects, findings, etc. Referring to comment 2 and this comment, as the descriptors and the presentation is not clear, the discussion is not convincing. Page 20, how the life diagram is developed is not clear and the linkage with the findings of this study and the discussions in previous sections should be strengthened.
2.
Section 2 must be improved. Which standard guidelines is being followed? The PRISMA or other standard protocol should be considered. Are the general database platforms for journal articles also researched, with currently considerable portion of studies referenced from unpublished work? Search strings and other validity details are not mentioned. What are the inclusion and exclusion criteria?
3.
The introduction from page 1-4 (ln 24 – 172) should be more precise. There is need to reduce redundancy in order to make a strong and specific introduction which concisely discusses the relevant background and the significance and rationales of the current study. Editing issues are also concern. For example, are Section 1.3 and Section 1.5 missing? Or is the numbering of sections incorrect? Please refer to ln 64 and ln 120. In addition, there should be further check and refinement of writing on the whole article. For example, ln 78, the word “Youn” should be revised as “Young”.
4.
Recommendations should be further derived from the discussions of the included literature and further developed into more details, instead of the given short section on page 21, ln 790 – 798.
5.
Accurate referencing should be addressed. For example, regarding the in-text citation of Shih & Wu (2017), is it referring to reference item 51 (page 23), which is cited as Shih, Y.H. (2017); or is it referring to reference item 55 (page 24), which is cited as Shih, Y.H., & Wu, P.F. (2018)?
The in-text citations are adopting author-name style but there are some incorrect sequence of reference items observed in the references section (p. 21-24) which made the readers difficult for tracing. For example, item 26 Faulconbridge et al. (2018) should be placed before items 16 – 25. Item 24 (Huang 2004) and item 19 (Huang et al. 2011) should be in consecutive order?
Inconsistency should be addressed. For example, page 7 (table 1) and ln 295, the work by Lai is cited with the year 2014, in ln 311, the work by Lai is cited with the year 2013. In the references section on page 23, only one item of Lai is found, which is cited with the year 2014. The author names in the in-text citation (ln 323) are different with the one in item 70 (ln 979), one is indicated as Ye while the other is displayed as Yeh.
Ln 386-393, I cannot trace the cited reference (Li, 2015) when reading the descriptions in this paragraph. There is no corresponding reference item of Li (2015) in the references section (p. 23). Is the author referring to item 29, the study in central Taiwan? Similarly for the cited reference in ln 394 regarding Li (2015).
Some other examples are also observed. In ln 683 and 686, the respective in-text references by Chen et al. 2012 and Chang 2006 are also not found in the references section. Please check all the references again.